# Implementing Innovative Approaches to Improve Health Care Delivery Systems for Integrating Communicable and Non-Communicable Diseases Using Tuberculosis and Diabetes as a Model in Tanzania

**DOI:** 10.3390/ijerph20176670

**Published:** 2023-08-29

**Authors:** Stellah G. Mpagama, Kenneth C. Byashalira, Nyasatu G. Chamba, Scott K. Heysell, Mohamed Z. Alimohamed, Pendomartha J. Shayo, Albino Kalolo, Anna M. Chongolo, Catherine G. Gitige, Blandina T. Mmbaga, Nyanda E. Ntinginya, Jan-Willem C. Alffenaar, Ib C. Bygbjerg, Troels Lillebaek, Dirk L. Christensen, Kaushik L. Ramaiya

**Affiliations:** 1Kibong’oto Infectious Diseases Hospital, Mae Street, Lomakaa Road, Siha Kilimanjaro 25401, Tanzania; kbyashalira1@gmail.com (K.C.B.); pendojs@gmail.com (P.J.S.); annachongolo3@gmail.com (A.M.C.); cathygitige@gmail.com (C.G.G.); 2Kilimanjaro Clinical Research Institute, Kilimanjaro Christian Medical University College, Moshi Kilimanjaro 25116, Tanzania; nyasatuchamba@yahoo.com (N.G.C.); b.mmbaga@kcri.ac.tz (B.T.M.); 3Kilimanjaro Clinical Research Institute, Moshi Kilimanjaro 25116, Tanzania; 4Division of Infectious Diseases and International Health, University of Virginia, Charlottesville, VA 22908-1340, USA; skh8r@uvahealth.org; 5Department of Internal Medicine, Hindu Manda Hospital, Ilala, Dar es Salaam 11104, Tanzania; mzahir@blood.ac.tz (M.Z.A.); ceo@hc.shm.or.tz (K.L.R.); 6Department of Haematology and Blood Transfusion, Muhimbili University of Health and Allied Sciences, Dar es Salaam 11103, Tanzania; 7Department of Public Health, Faculty of Medicine, St. Francis University College of Health and Allied Sciences, Ifakara 67501, Tanzania; kaloloa@gmail.com; 8National Institute of Medical Research-Mbeya Medical Research Centre, Hospital Hill Road, Mbeya 53110, Tanzania; nelias@nimr-mmrc.org; 9Faculty of Medicine and Health, School of Pharmacy, University of Sydney, Sydney, NSW 2006, Australia; johannes.alffenaar@sydney.edu.au; 10Sydney Institute for Infectious Diseases, University of Sydney, Sydney, NSW 2145, Australia; 11Westmead Hospital, Sydney, NSW 2145, Australia; 12Global Health Section, Department of Public Health, University of Copenhagen, DK-1353 Copenhagen, Denmark; iby@sund.ku.dk (I.C.B.); trli@sund.ku.dk (T.L.); dirklc@sund.ku.dk (D.L.C.); 13International Reference Laboratory of Mycobacteriology, Statens Serum Institut, DK-2300 Copenhagen, Denmark

**Keywords:** integration, communicable and non-communicable diseases, tuberculosis, diabetes

## Abstract

*Background:* Many evidence-based health interventions, particularly in low-income settings, have failed to deliver the expected impact. We designed an Adaptive Diseases Control Expert Programme in Tanzania (ADEPT) to address systemic challenges in health care delivery and examined the feasibility, acceptability and effectiveness of the model using tuberculosis (TB) and diabetes mellitus (DM) as a prototype. *Methods*: This was an effectiveness-implementation hybrid type-3 design that was implemented in Dar es Salaam, Iringa and Kilimanjaro regions. The strategy included a stepwise training approach with web-based platforms adapting the Gibbs’ reflective cycle. Health facilities with TB services were supplemented with DM diagnostics, including glycated haemoglobin A1c (HbA1c). The clinical audit was deployed as a measure of fidelity. Retrospective and cross-sectional designs were used to assess the fidelity, acceptability and feasibility of the model. *Results*: From 2019–2021, the clinical audit showed that ADEPT intervention health facilities more often identified median 8 (IQR 6–19) individuals with dual TB and DM, compared with control health facilities, median of 1 (IQR 0–3) (*p* = 0.02). Likewise, the clinical utility of HbA1c on intervention sites was 63% (IQR:35–75%) in TB/DM individuals compared to none in the control sites at all levels, whereas other components of the standard of clinical management of patients with dual TB and DM did not significantly differ. The health facilities showed no difference in screening for additional comorbidities such as hypertension and malnutrition. The stepwise training enrolled a total of 46 nurse officers and medical doctors/specialists for web-based training and 40 (87%) attended the workshop. Thirty-one (67%), 18 nurse officers and 13 medical doctors/specialists, implemented the second step of training others and yielded a total of 519 additional front-line health care workers trained: 371 nurses and 148 clinicians. Overall, the ADEPT model was scored as feasible by metrics applied to both front-line health care providers and health facilities. *Conclusions*: It was feasible to use a stepwise training and clinical audit to support the integration of TB and DM management and it was largely acceptable and effective in differing regions within Tanzania. When adapted in the Tanzania health system context, the model will likely improve quality of services.

## 1. Background

Although in the last decade there have been considerable advances in science, technologies and innovations to address major infectious diseases such as tuberculosis (TB) and human immunodeficiency virus (HIV) in Low- and Middle-Income Countries (LMICs), the projected impact in improving the quality of services has not been realized [1]. Poor quality of services in LMICs is responsible for up to 5 million individual deaths per year, approximately 15% of overall deaths, which can be considered an epidemic unto itself [2,3].

Health systems in sub-Saharan African (SSA) struggle not only to deliver quality services to individuals affected with TB and HIV, but the region is also observing the concurrent rise of non-communicable diseases (NCDs) such as diabetes mellitus (DM), hypertension and chronic lung diseases. Communicable diseases and NCDs are usually managed in separate sections of the health system, but those sections can be equally susceptible to emerging economic and biosecurity concerns such as COVID-19, which further challenge already fragile health systems [4,5].

In Tanzania, 75,000–85,000 TB cases are officially added to health system records annually with 20–30% of those people co-infected with HIV. However, one of the other most common drivers of TB disease is DM, prevalent in 9–16% of those with TB in the country and often underdiagnosed [6,7,8]. The global response toward TB and HIV has led to equally momentous changes in recommendations and guidelines such as the lipoarabinomannan diagnostic test or novel TB preventive therapy. Countries like Tanzania still face systemic challenges in effective delivery of these updated recommendations to front-line health care workers with adequate support for implementation [9,10]. Such recommendations and guidelines also suggest the benefit of integrating services for NCDs with TB and HIV care, but this has not been accomplished in Tanzania [11]. For instance, the introduction of molecular diagnostics for multidrug resistant (MDR)-TB in Tanzania did not translate into a reduction of mortality as described elsewhere [12], prompting a nationwide examination of barriers and bottlenecks [13]. Importantly, this examination found that most of the front-line health care providers in TB and HIV clinics did not regularly receive continuous on-the-job medical education and therefore lacked updated knowledge and skills on the international standards of TB care. For example, 83% of front-line health care workers in TB and HIV clinics had unacceptable proficiency regarding the clinical application of molecular diagnostics as endorsed by the World Health Organization (WHO) [10]. As a consequence, a mere 30% of patients with presumed MDR-TB were able to access the recommended diagnostics for optimal clinical management [10]. Some of the common challenges that contributed to under-implementation include lack of awareness and training with the novel tests and pathways for acquisition of consumables necessary for use [10].

Although evidence suggests that integration of infectious diseases and NCDs may improve the number of people receiving health care, this may not necessarily, improve the quality-of-service delivery or the ultimate health status of individuals accessing care [14]. Therefore, we designed an intervention for altering health care delivery with three interdependent domains: (i) a stepwise training approach for knowledge and skills acquisition for front-line health care providers, (ii) adaptive service delivery through integration of communicable and NCDs using TB and DM as a case study, and (iii) continuous learning and integration of dual communicable and NCDs as described elsewhere [15]. We adopted a theory of change such that this system would be self-repairing and subsequently shift health care delivery systems to a more patient-centred focus [16]. We examined the feasibility, acceptability and effectiveness of the model, estimated the extent of the integration of practice for patients with dual TB/DM disease, and measured relevant individual health outcomes using a clinical audit tool. 

## 2. Materials and Methods

### 2.1. Method Description

#### 2.1.1. Design

This was an effectiveness-implementation hybrid design type-3 strategy described as testing an implementation strategy while gathering information on clinical interventions and outcome. The design was implemented while observing the integration’s effect on the bidirectional screening of TB and DM in patients with or without HIV [17]. Infection prevention control (IPC) was core for TB/DM services and “one stop shops” were proposed in TB clinics only. To prevent the possible transmission of TB to other patients attending DM clinics, this approach was not practiced in those DM clinics. DM clinics were required to refer the diagnosed TB/DM cases to TB clinics where IPC protocols were already in place [18].

The ADEPT model was implemented simultaneously in health facilities and compared, with others not implementing within the same region labelled as intervention and control accordingly. The implementation process started in September 2019 with an initial phase of training of graduate nurses and doctors in three selected regions, namely, Dar es Salaam, Iringa and Kilimanjaro with a comprehensive educational and services package for TB and DM with or without HIV. The second phase was a cross-sectional design that examined the feasibility of the process through assessing the performance percentage scores of the trained health care providers and health facilities implementing integration of TB and DM management using pre-defined criteria (Appendix A). Unfortunately, phase 2 was interrupted with lockdown measures due to the COVID-19 pandemic that reached the country in early 2020 [19]. The third phase was also a cross-sectional design and included the clinical audit of the selected clinical standards for dual TB and DM services in each of the participating health facilities. This study complied with the Declaration of Helsinki and obtained ethical approval from the local ethical clearance with reference KNCHREC003 and the National Health Research Ethical Committee (NIMR/HQ/R.8a/Vol.IX/2988). This manuscript is reported according to the standards for reporting implementation studies [20].

#### 2.1.2. Context

The ADEPT implementation included three regions—Dar es Salaam, Iringa and Kilimanjaro. Prevalence of DM in the general population in Dar es Salaam and Kilimanjaro is 9% and 5.7%, while the prevalence of DM in people with TB is estimated at 9.7% and 9.2%, respectively [7,21,22]. Studies of the magnitude of DM in the general population in Iringa were lacking, but one study found a prevalence of DM among people with TB of approximately 9% [23]. Dar es Salaam and Iringa also have a high proportion of TB and HIV coinfection [21,24]. In each region, health facilities offering services in rural, semi-urban and urban settings were included.

#### 2.1.3. Target Sites and Sample Size

The target sites were three levels of health facilities including regional referral hospitals, district hospitals and health centres/dispensaries owned by either the government or faith-based organizations, or private facilities that provided either TB or DM services. We purposefully selected 40 total health facilities, including at least 10 health facilities from each region, as recommended elsewhere [25]. Each health facility contributed at least 1 nurse officer and 1 medical officer/specialist. The enrolled participants for the initial step included regional and district coordinators for TB and NCDs for health system harmonization [15].

#### 2.1.4. Description of the ADEPT Model Strategy

The ADEPT model is a multifaceted interventions package composed of adoption of digital technologies for acquiring knowledge and workshops for enhancing practical skills. In addition, the model engaged medical specialists, particularly internal medicine postgraduate practitioners, for championing integrations of communicable disease and NCD at the health facilities. The programme was implemented in collaboration with the Ministry of Health and the President Office Regional Administration and Local Government Authority. The programme engaged Regional and District Medical Officers in selecting trainees.

Trainees were stratified into two clusters; the first cluster included graduate nurses or doctors/medical specialists working in general or specific clinics related to TB/HIV or DM, and the second cluster included nurses and clinicians at all levels of certification who were trained by the first clusters, thereby expanding the number of health care workers within and outside the health facilities. Likewise, the ADEPT-implementing health facilities received diagnostics for DM services and knowledge for acquisition and stock management. This was not the case for the control health facilities. Diagnostics distributed included the point-of-care glucometer machine (GlucoPlusTM Inc. 2323 Halpern, Ville St-Laurent, Québec, Canada), and HbA1c analyser (HemoCue Hb1c 501 system-HemoCue AB; SE-262 23, Ängelholm, Sweden), whereas TB diagnostics were assessed as commonly available. The clinical audit was introduced semi-annually as an auto-learning mechanism that guided modifications and improvement of the clinical practice when given feedback indicating inconsistent or sub-optimal implementation of the clinical standard. Figure 1 describes the logic model conceptualized theory of change. The ADEPT implementation strategy was organized in three series phases summarized here.

#### 2.1.5. Phase 1: Training of Cluster One of the Health Care Providers

Training of the front-line health care providers exposed graduate nurses and doctors/specialists to a web-based training for acquiring knowledge on TB and DM and associated comorbidities such as HIV and hypertension, respectively. The second phase of training was to participate in a one-week workshop for acquiring skills to utilise technologies and other hands-on skills in TB and DM. Facilitators used Gibb’s reflective cycle to reinforce knowledge acquisition of TB, DM, HIV, related diseases, diagnostics, clinical management and integration with differential diagnoses from experience and integrating proposition, professional craft and personal knowledge [26]. Failure to participate or meet the criteria set was a disqualification for the next step. Repeated training could be taken for meeting the criteria. The important aspects of training were the clinical standards of bidirectional screening TB and DM and clinical management of patients with dual diseases [27]. This apprenticed learning through the full web-based and workshop training was followed by materials for not only integrating dual TB and DM at their respective health facilities but also instructions to guide other health care providers at different levels of certification.

#### 2.1.6. Phase 2: Training of Level 2 and Integration of the TB and DM Services 

Following the qualified health care providers’ (from step 1) return to their health facility, they trained other health care providers (clinicians and nurses)—cluster 2. The training was directive and emphasized hands-on implementation of the bidirectional screening and clinical management of TB and DM. Assessment of the performance of the service was planned to be 3 months after qualification. 

Performance was assessed in the utilization of the DM screening algorithm in people recently diagnosed with TB. The algorithm first used a glucometer, and then to exclude patients with transient hyperglycaemia due to cytokine stimulation (false DM diagnoses) [28], the algorithm used Hb1Ac to confirm the diagnosis and assess the severity of DM while guiding the urgency of clinical management of DM in TB cases. Both glucometer and HbA1c result interpretation followed the national and WHO guidelines and the treatment of patients with dual TB and DM followed the current guidelines [29].

Concurrently, people with new or previously diagnosed DM were also screened for active TB. Key questions, similar to those being asked about intensified TB case findings in people living with HIV, were applied. The questions included a history of cough for at least 2 weeks, haemoptysis, night sweats, fever, weight loss, chest pain and difficulty in breathing or chest tightness. However, in poorly controlled diabetes, classical symptoms may be altered [29]. A standard laboratory test algorithm for a presumed TB case was followed. The algorithm included collection of sputum for light emitted diode (LED) microscopy or XpertMTB/RIF for facilities that were already utilizing XpertMTB/RIF.

Furthermore, trainers in cluster 1 were asked to review the infection control measures in their respective health facilities and design a “one stop shop” or an alternate clinic day for TB/DM to prevent possible TB transmission to individuals with DM. Subsequent infection control policies and the methods for visit days for TB/DM dual care were therefore also assessed. Lastly, referral and linkages to specialized care through an expert TB/DM panel and a direct consultation liaison were organized to provide distant support in managing complex TB or DM cases while allowing the patient to remain connected to their primary health care practitioners and clinics. Patients did not receive any monetary incentive. The criteria used for assessing mentors and health facilities are available in Appendix A.

#### 2.1.7. Phase 3: Clinical Audit of the TB and DM Clinical Standards

We then introduced a clinical audit practice to assess the quality and implementation of the clinical standards for dual integration of TB and DM services particularly on procedures of bidirectional screening and clinical management of patients with dual TB/DM [29]. This stage aimed to evaluate a one-year clinical practice. Doctors/medical specialists conducted the clinical audit following the stipulated clinical standards. Clinical auditors were not allowed to assess their home health facility to avoid bias. Clinical audit tools are attached in Appendix A.

### 2.2. Method Evaluation

The primary outcome was the percentage score of the implementation of the bidirectional screening of TB or DM regardless of HIV status in routine clinical settings as estimated using the clinical audit measure (Appendix A). This was calculated as the proportion of individuals appropriately exposed to the clinical standard to the total number of patients with a condition attended at the health facility. The reference for the clinical standards for the bidirectional screening of patients with TB and DM and appropriate clinical management was based on the national guideline [24]. Secondary outcomes included the proportion of TB or DM patients screened with other comorbidities such as hypertension and anthropometry, as well as the proportion of health care workers enrolled and able to deliver training in cluster 1 and the total number of cluster 2 trainees. Moreover, the feasibility of integrating dual TB and DM screening and management was appraised using the task allocated as summarized in the tools designed for cluster 1 health care workers and health facilities (Appendix A). Furthermore, health facilities’ evaluable tasks included awareness of health facility leaders and workers, organization and appropriate clinical management of TB/DM including linkages addressing comorbidities and pharmacovigilance. Each parameter had an estimated score and calculations of each parameter were estimated with a total percentage score estimating the achievement of cluster 1 (nurses and medical specialists) and health facilities. Acceptability evaluation of the stepwise approach was assessed among front-line health care workers (cluster 1 and 2) using a Likert scale and evaluated parameters included clarity of course objectives, quality of text and images, whether the course was engaging, and if it changed own practices and recommended others to change. Field implementation challenges described by the front-line health care workers were documented and summarized during field visits.

#### Statistical Analysis

Health facilities were categorized as hospitals if they served the district and above or health centres if they served below the district level. Categorical data were presented in proportions with percentages whereas all numerical data were skewed; thus, median with interquartile range (IQR) between 25 and 75 estimated the measures of central tendencies and dispersion. Chi-squared tests were used to compare the categorical variables and Mann–Whitney U tests were used for continuous variables. All statistical tests were two-tailed, with a *p*-value < 0.05 considered significant. Data were entered into Microsoft Excel (Version 16.59, Microsoft Cooperation, Redmond, WA, USA, and were analysed using SPSS (Version 25, SPSS Inc, Chicago, IL, USA).

## 3. Results

In 2021, 40 health facilities were assessed for their performance during the entire year of 2020. Unfortunately, COVID-19 waves interrupted completion of the processes and only 25 (63%) facilities were evaluated. The evaluable ADEPT model intervention health facilities and control facilities were 14 (56%) and 11 (44%), respectively. Nine (36%) facilities were hospitals and 16 (64%) were health centres. There was no difference in the distribution of intervention hospitals 5 (56%), health centres 9 (56%) and one-to-one controls (*p* = 0.65). At the hospital level, the median (IQR) number of individuals diagnosed with TB in year 2020 at the intervention facilities was 520 (315–501) while control was 320 (314–520, *p* = 0.73). Likewise, the median (IQR) number of individuals receiving DM services in year 2020 at the intervention hospitals was 1760 (309–2570) and control was 98 (30–6279, *p* = 0.89).

At the health centre level, the median (IQR) individuals with TB diagnosed in the year 2020 at the intervention facilities was 330 (42–524) and control facilities was 330 (38–557) (*p* = 0.29), whereas the median (IQR) individuals with DM attending the services in the same time period was 160 (0–757) in intervention facilities and 60 (0–1598) control health centres (*p* = 0.89). Individuals were more often identified with dual TB and DM at the intervention hospitals with a median of eight (6–19) individuals per facility compared with a median of one individual per control facility 1 (0–3) (*p* = 0.02). The same trend was observed in health centres; dual TB and DM was diagnosed in a median of four individuals (4–5) per facility in intervention facilities compared to none at any of the control facilities (*p* = 0.01).

### 3.1. Stepwise Training ADEPT Intervention Sites

From July to September 2019, 46 nurse officers and medical doctors/specialists that served at the general clinics or specific clinics for TB or DM or internal medicine were recruited for web-based training and 40 (87%) attended the workshop. Thirty-one (67%), 18 nurse officers and 13 medical doctors/specialists, implemented the second step of training. Reasons for attrition are described in Figure 2. The first cluster diffused knowledge and skills to other health care workers, which between September and December 2019 totalled an additional 519 individual health care providers, 371 nurses and 148 other clinicians. Within the compounding model, for every one nurse in cluster 1, another 21 nurses on average were trained in cluster 2, and for every one clinician in cluster 1, another 11 clinicians were trained in cluster 2.

### 3.2. Comparison of Outcome Measures Using Clinical Audit Tools in Intervention and Control Health Facilities

Compared to control health facilities at both hospital and health centre levels, the clinical standards for bidirectional screening of TB and DM were implemented significantly more often in intervention health facilities. However, health centres more often implemented the screening of TB in DM clinics compared to the hospitals as shown in Table 1. Although the total score on implementation of the standards at the TB clinics was significantly high, this was not the case in the DM clinics. Furthermore, in either TB or DM clinics, there was no difference in the performance of the intervention and control health facilities for screening of additional comorbidities such as hypertension and malnutrition as summarized in Table 1. Despite these trends, the total performance of the seven components itemized in Table 1 of the standard for optimal clinical management of patients with dual TB and DM was significantly better in intervention facilities compared to the control facilities.

### 3.3. Feasibility of the Stepwise Training

In the first three months, the feasibility assessment of the stepwise training in establishing integration of the dual TB and DM and effect in clinical care were assessed using task allocated tools at various health facility level and cadres. The mean percentage score of the tasks that cluster 1 achieved across all three regions for nurses was 62.2% whereas for medical doctors/specialists it was 55%, and for health facility scores: hospitals were 64.8% and health centres were 69.7%. Stratification of health facilities by clustering 1 and 2 scores were 76.2% and 57.1%, respectively. Except for the nurses in Iringa region who outperformed (76.3%) those in Dar es Salaam (65%) and Kilimanjaro (54%) (*p* = 0.001), other task achievement was similarly achieved across the three regions as shown in Table 2.

### 3.4. Acceptability of Web-Based Training in the Stepwise Training

All participants agreed on the use of web-based training as summarized in Figure 3.

Recommendations from participants on the courses that may be included in future web-based trainings included drug-resistant TB, atypical mycobacterial disease, diabetes retinopathy, renal diseases, gastrointestinal infections, sexually transmitted infections, critical care and cancers.

Furthermore, 79 (15%) of the cluster 2 participants were assessed from their health facilities and 72 (95%) were either very satisfied or satisfied with cluster 1 trainees visiting their health facilities to train them on the dual TB and DM clinical management. Only 5% were dissatisfied and the described reasons included the time limit, and the fact that the mentorship was provided during working hours and that participation was not effective.

### 3.5. Other Observed Challenges during the Process of Implementation

Recording and reporting tools were available only for TB but not for tracking the integration of dual TB and DM and other associated comorbidities such as hypertension or malnutrition. Informal registries were designed to capture some data and created duplications of efforts as shown in Figure 4. Example of registries that health facilities used for recording the screening of DM in TB and clinical management of patients with dual TB/DM are shown in the picture. Likewise, there were no reporting mechanisms from the health facilities to the higher levels of the health system for prioritization of services, logistics, training of front-line health care providers, or procurement of diagnostics or drugs for TB and DM and other associated comorbidities.

## 4. Discussion

Health facilities that participated in the ADEPT intervention detected substantially more individuals with dual TB and DM in both TB-dedicated clinics and DM-dedicated clinics compared to control facilities. Management of dual TB/DM in intervention facilities was also more likely to reflect standards and skills acquired by health care providers participating in cluster 1. The ADEPT model of compounding training to new health care providers allowed for rapid dissemination of skills to utilise the applied diagnostic tests and algorithms. Furthermore, the process of self-audit (health care workers audit another peer health facility) created accountability mechanisms that will subsequently increase the proportions of individuals at all clinics receiving evaluations and appropriate clinical management according to the defined clinical standards [30]. In other settings, audit data have raised issues for dialogue among front-line health care providers and leaders and legitimized provision of feedback to colleagues while encouraging supportive collaborations [31]. The ADEPT model also backstopped resources and necessary expertise for the audit process, without which other audit processes have failed [32]. 

We also found that in addition to the feasibility of stepwise training to integrate TB and DM dual screening and management at scale, the ADEPT model also empowered collective leadership among front-line health care workers through expanding a pool of graduates with shared common knowledge and the means and purpose to communicate with one another. This is a modified approach of the train-the-trainer model that considers mentorship in the workplace and evaluation of the field practice [33]. Unexpectedly, cluster 2 health facilities in Dar es Salaam outperformed all the health facilities in the appropriate implementation of the bidirectional screening of TB and DM and in delivery of appropriate dual TB and DM services. Yet importantly, there was no difference in the performance percentage score between health provider facilities from cluster 1 and those from cluster 2, suggesting that the training that another health care provider can provide to a peer in this setting is durable. Further training and retraining will likely raise the proportion of those completing the training, particularly in areas with high attrition. 

Favorably, the front-line health care providers accepted the use of the web-based platform and provided favourable comments such as expanding the scope of modules to include other medical conditions relevant to the daily practices. Similar results were also observed in a large SSA region-wide survey on the use of web-based continuous medical education [29]. The approach is novel for the setting as modules may be purposefully short and directive in scope but additive, so that health care providers can access the content in their own time, at their own place and at their own pace. This form of web-based training also has the potential to link with professional nursing or medical recertification. We predict that this form of educational intervention for all front-line health care providers would otherwise preclude required enormous mobilized funding, trainers and venues [34].

One of the challenges observed in this study was lack of integration of health care diagnostics and management algorithms with health management information systems. This is a major impediment in SSA where frameworks for health information system are often donor-dependent, and the majority of countries lack a national strategy [35]. As a result, data were difficult to harmonize and this could not have been accomplished without the external support for this project. As such, the ADEPT model for management of TB and DM dual disease is not yet fully sustainable, and will require additional external investment in health information systems to capture data from currently fixed sources. Despite the lack of formal pre-existing NCD recording tools, health care providers innovated informal registers, and while these could be considered for further use, they ultimately created duplication of effort. As many health facilities used informal data recording tools, results were not directly shared with the higher authorities. Failure of the health facilities to inform authorities that could enact changes in budgetary scope or resource acquisition likely led to a lack of problem visibility and subsequent inaction or continuance of the standard of practice. 

Other potential limitations in interpreting the generalizability of the findings include the selection of health facilities without randomization to the intervention. The total number of health facilities selected, the distribution of intervention and control sites and the minimum of 10 sites in each region likely minimized any confounding by assignment. In addition, there were various ongoing global health intervention initiatives in Tanzania that might have increased the capacity of certain systems for health care delivery introducing bias in the results; but similarly, this was likely offset by our diverse recruitment of many health facilities at various levels and settings [36]. Even though COVID-19 waves interfered with the evaluation processes, with some health facilities therefore not being assessed in the clinical audit phase, this lack of evaluation was distributed equally across regions and types of health facilities, likely minimizing bias. 

## 5. Conclusions

Implementation of the ADEPT intervention intended to integrate communicable and non-communicable diseases, yet it was partially interrupted by the emergence of COVID-19. However, as the pandemic exposed longstanding cracks in public health infrastructure [37], the ADEPT model became all the more relevant in providing an example of nimble and efficient service integration [38]. Indeed, conservative modelling has demonstrated that COVID-19 related disruption of health services will result in an excess of death from HIV of up 10%, from TB of up to 20%, and malaria of up to 36% over the next 5 years in high-burden settings [39]. Likewise, people living with DM are both at direct increased risk of severe disease and death from COVID-19, and indirect risk from disruption of health services, even in high-income countries [40]. The time for bold strategies such as ADEPT is now.

## Figures and Tables

**Figure 1 ijerph-20-06670-f001:**
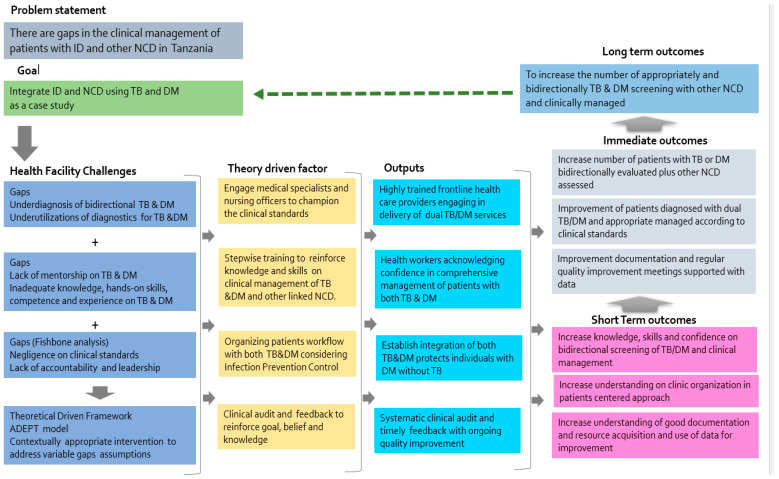
LEGEND. ADEPT framework and strategy included gap analysis that combined the views and thinking of the front-line health care providers through fish bone analysis. The driven design of the theory of change included stepwise training and clinical audit. Outputs expected to improve knowledge and skills to reinforce initiation of the integration of dual TB and DM services and other associated comorbidities. This will subsequently strengthen integration of ID and NCD in health care delivery systems.

**Figure 2 ijerph-20-06670-f002:**
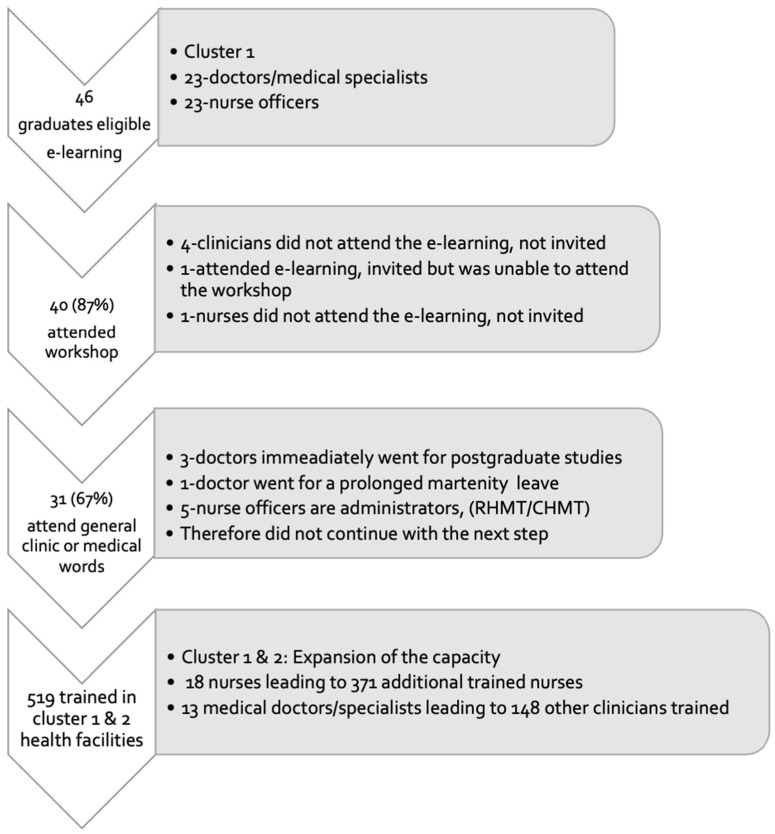
LEGEND. Cluster 1 (mentors) exposed to multiple steps including the web-based training for acquiring knowledge, workshop for acquiring skills on technologies and hands-on skills in the field. Failure to participate in one of the steps disqualified the mentorship role. RHMT-Regional Health Management Team. CHMT-Council Health Management Team.

**Figure 3 ijerph-20-06670-f003:**
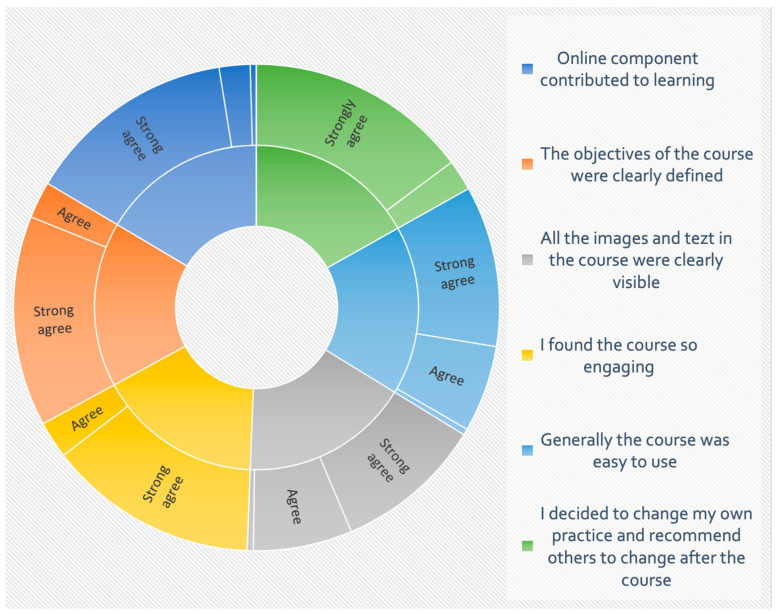
LEGEND. Forty participants evaluated the online course delivery. Participants strongly agree on the assessed parameters by >98%. Strong disagree and disagree was recorded in 1 (2.5%) and 1 (2.5%) on whether the online component contributed to learning and if all the images and text in the course were clearly visible correspondingly. The course was highly acceptable to the participants.

**Figure 4 ijerph-20-06670-f004:**
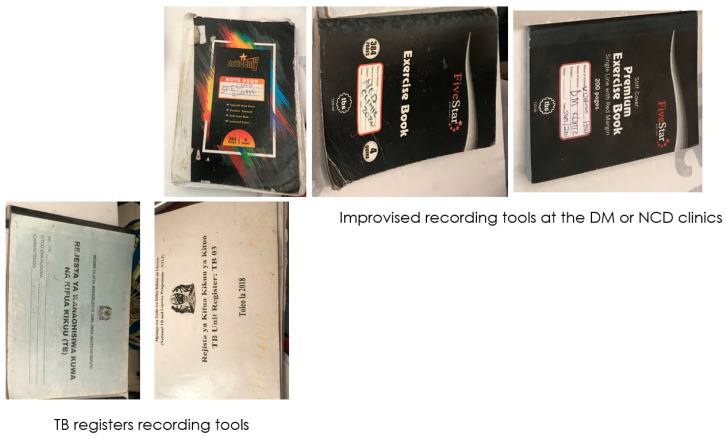
LEGEND. Recording and reporting tools observed at the clinics participating in ADEPT intervention. The TB programme has well-designed recording and reporting tools contrary to DM. Likewise, there are no tools for reporting integrated services. Decision makers will not access information on the burden and needs of dual TB and DM.

**Table 1 ijerph-20-06670-t001:** Implementation fidelity of the intervention as shown by the clinical audit findings.

Characteristic	Sub-Category	Total	Intervention Health Facilities Median Percentage Score in n% (25/75) *	Control Health Facilities Median Score in n% (25/75)	*p*-Value
**Standard # 1: All new TB patients should be screened for diabetes (DM) at the start of TB treatment using the DM screening questionnaire to identify those with symptoms and signs of DM. Random blood glucose (RBG) and fasting blood glucose (FBG) tests should be performed per the algorithm for diagnosis of DM among TB patients.**					
DM screening in TB, median (IQR)	Hospitals	10 (0–69)	90 (60–100)	0	0.00
	Health Centres	25 (0–95)	69 (35–70)	0	
Hypertension screening in TB, median (IQR)	Hospitals	0	0	0	0.24
	Health Centres	0	0 (0–10)	0	
Malnutrition screening in TB median (IQR)	Hospitals	23(10–30)	25 (23–30)	14 (5–28)	0.31
	Health Centres	12 (5–27)	5 (5–30)	15 (4–25)	
General implementation of the standard, median (IQR)	Hospitals	23 (8–46)	46 (45–48)	7 (3–8)	0.00
	Health Centre	10 (2–52)	48 (13–62)	2 (0–6)	
**Standard # 2: All DM patients should be screened for TB at the time of diagnosis and at follow-up visits.**					
TB screening in DM, median (IQR)	Hospitals	0 (0–67)	0 (0–69)	0 (0–50)	0.05
	Health Centres	0 (0–75)	75(29–100)	0	
Hypertension screening in DM, median (IQR)	Hospitals	10 (0–100)	0 (0–10)	100 (50–100)	0.81
	Health Centres	0 (0–12)	12 (0–95)	0	
Malnutrition screening in DM, median (IQR)	Hospitals	0	0	34(0–84)	0.86
	Health Centres	0	0 (0–3)	0	
General implementation of the standard, median (IQR)	Hospitals	19 (0–32)	0 (0–19)	39 (12–56)	0.60
	Health Centres	0 (0–19)	19 (14–32)	0	
**Standard # 3: Management of patient with dual TB and DM either co-infected with HIV or not; the treatment will follow the standard TB treatment guideline and DM management will be in accordance with HbA1c glycaemic levels.**					
HbA1c testing	Hospital	27 (0–50)	45 (27–50)	0	0.04
	Health centre	80 (42–100)	80 (42–100)	0	
Clinical management according to HbA1c results and antiretroviral therapy	Hospital	45 (38–50)	45 (41–58)	25 (0–50)	0.55
	Health centre	50 (0–82)	50 (0–82)	0	
Assessment of complications	Hospital	0 (0–20)	20 (0–58)	0	0.19
	Health centre	0 (0–31)	0 (0–35)	0	
Management of comorbidities	Hospital	0 (0–58)	29 (0–69)	0	0.35
	Health centre	25 (0–71)	25 (0–71)	0	
Recording and reporting adverse drug reactions	Hospital	0	0 (0–4)	0	0.35
	Health centre	8 (0–100)	8 (0–100)	0	
Linkage to the DM or DM/HIV clinic after TB treatment	Hospital	0(0–80)	40 (0–90)	0	0.19
	Health centre	100 (50–100)	100 (50–100)	0	
General implementation of the standard, median (IQR)	Hospital	25 (8–38)	32 (25–46)	4 (0–8)	0.04
	Health centre	49 (22–70)	49 (22–70)	0	

* Total number of individuals screened divided by the total number of individuals at the health facility with TB (standard 1) DM (standard 2) and dual TB/DM (standard 3).

**Table 2 ijerph-20-06670-t002:** Performance of health care providers and facilities in the first 3 months of initiating ADEPT interventions.

Characteristics	Subcategory	All n%	Dar n%	Iringa n%	Kilimanjaro n%	*p*-Value
Health care provider scores ^1^	Nurses	65.2	65.0	76.3	54.0	0.001
	Doctors/specialists	55.0	67.2	49.0	55.0	0.665
Health facility score ^2^	Cluster 1	76.2	61.9	73.3	76.2	0.966
	Cluster 2	57.1	81.0	53.3	57.1	0.091
Type of health facility score	Hospitals	64.8	74.6	58.7	61.9	0.294
	Health centres	69.7	72.2	68.6	69.2	0.795

^1^ Health care provider scores for nurses and doctors/medical specialists included the percentage achievement in empowering other health care workers at the workplace and distant facilities, demonstrating hands-on pharmacovigilance practices and infection prevention control. Specifically for nurses, it included achieving the minimum target of the quality of nursing care plans for the patients with TB and associated comorbidities whereas for doctors/medical specialists, it included bidirectional screening of TB and DM and appropriate clinical management including other comorbidities such as hypertension and utilisation of HbA1c. ^2^ Health facility score combines the extent of achievement including awareness and commitment of the managers, organisation and clinical management of patients with dual TB and DM, established referral mechanisms and health educations, including dual TB and DM. Furthermore, health facilities were categorised into hospitals and health centres.

## Data Availability

Datasets used during this study are available from the corresponding author upon reasonable request.

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
