# Peer review of "Implementing Innovative Approaches to Improve Health Care Delivery Systems for Integrating Communicable and Non-Communicable Diseases Using Tuberculosis and Diabetes as a Model in Tanzania"

_ijerph, 2023, doi:10.3390/ijerph20176670_

Round 1
Author Response
- This is a well-conducted and well-written study to address the important health service needs. It was conducted in a difficult time due to COVID-19 epidemic. Even though with the challenge and difficulty, it confirms that using a stepwise training and clinical audit to support the integration of TB and DM management was feasible. Findings from this research shows that it was largely acceptable and effective in differing regions within Tanzania. Findings of this study partially fill a knowledge gap and will contribute with important information on health service needs. Just some minor issues and suggestions to improve quality and easy reading for reader from outside of the country.
Reviewer #1 response
Thank you very much the reviewer for compliment and acknowledging the importance of this research subject. We appreciate for your time invested in thorough review, comments and suggestion for improving this report. Below, you will find a point-by-point description of how the reviewer comments have been attended in the manuscript. Original reviewer’s comments are in regular face, responses in red typeface.
- Line 112: Why infection control measure was proposed in TB clinics only? As descripted in line 122-125, the phase 1 and phase 2 training on integration…also involved in diabetes clinics if my understanding is correct. Please clarify.
Reviewer response
We proposed one stop shop model and offer TB/DM services and largely empower TB/DM care in TB clinics. The main reason is to prevent the possible transmission of TB to other patients attending DM clinics, because this approach was not practiced in those DM clinics. This justification is described on line 114 through 116. Likewise, all phases involved both TB and DM clinics. For DM clinics the component of active TB screening was highly promoted. If they identify a TB patient with DM, then will be managed preferably at the TB clinic for both diseases. This was important for patient safety as clinicians require to review drug-drug interaction and also prevent TB transmission from the index patient to another DM patient
- Line 101-192: It indicates that failure to participate or meet the criteria set was a disqualification for the next step. How many of them were finally failed to meet this qualification for next step? You may add it in the result.
Reviewer response
The methods section described conditions for meeting the criteria to qualify for next steps. The results section Line 293-8 described attrition at each step and figures are more shown on Figure2.
- Line 309-311: It mentioned that “Compared to control health facilities at both hospital and health centre levels, the clinical standards for bidirectional screening of TB and DM were implemented significantly more often in intervention health facilities.” In the previously paragraph, it also pointed out diagnosed more DM-TB cases in the intervention side. However, there is almost no discussion on this point. As this point has public health implication in term of increase case finding for both of the diseases, it should be sufficiently addressed in the discussion.
Reviewer response
Yes, indeed line 309-311 described the results. In this section, we have presented the results only and we did not blend with the discussion. The discussion of the findings is in the section titled discussion. Line 379-discussed findings that have been presented in Line 309-311
- Line 364-373: It mentioned some challenges from technical aspect. Did you encounter any challenge on shortage of staffing or too busy of frontline health workers that preventing them from participation to the training?
Reviewer response
We enquired information from staff the challenges they have encountered. This was not mentioned that is why is not included.
- Finally, I suggest to add a short paragraph of conclusions at the end of discussion.
Reviewer response
The last paragraph was a conclusion; thus, we have added a sub-title conclusion on that paragraph to address this comment but also meet the journal recommendations

Reviewer 2 Report
The overall quality of the paper is good. It is a well-written paper with interesting results and discussion. The overall presentation can be improved with careful proofreading, for example, the last bullet point in step 3 of Figure 2.
Minor errors in typing/grammar are present in the paper, for example, in the sentence (lines 56-57 )
there has been considerable advances in science, technologies and innovations to address major infectious diseases such as tuberculosis (TB)
The word has may be replaced with have been, a comma may be placed after technologies
In the sentence, The global response toward TB and HIV has led to equally momentous change in recommendations and guidelines such as lipoarabinomannan diagnostic test or novel TB preventive countries like Tanzania (lines 73-75)
Some text seems to be missing after the high lightened words above
The last bullet point in step 3 of Figure 2 seems to be part of the second-last sentence/bullet
Occasional Spaces missing between the text and references number at the end of the sentences, for example, personal knowledge(26).
Careful proofreading and correction with some online grammar apps can improve the quality of the paper
Author Response
- Comments and Suggestions for Authors
The overall quality of the paper is good. It is a well-written paper with interesting results and discussion.
Reviewer #2 response
We appreciate for compliment and for your time devoted in detailed review, comments and suggestion for improving this report. Below, you will find a point-by-point description of how we have attended the comments and suggestion in the manuscript. Original reviewer’s comments are in regular face, responses in red typeface.
- The overall presentation can be improved with careful proofreading, for example, the last bullet point in step 3 of Figure 2.
Reviewer response
Thank you for observations. We have done proofreading and corrected the errors for example “exapansion” now is corrected to expansion and also uncomplete statement was completed.
- Comments on the Quality of English Language
Minor errors in typing/grammar are present in the paper, for example, in the sentence (lines 56-57 )there has been considerable advances in science, technologies and innovations to address major infectious diseases such as tuberculosis (TB).The word has may be replaced with have been, a comma may be placed after technologies
Reviewer response
We have corrected the errors. Thank you
- In the sentence, The global response toward TB and HIV has led to equally momentous change in recommendations and guidelines such as lipoarabinomannan diagnostic test or novel TB preventive countries like Tanzania (lines 73-75). Some text seems to be missing after the high lightened words above
Reviewer response
Thank you for observation. We have added a missing word “therapy”. For clarity, we have split the sentence by putting full stop after therapy.
- The last bullet point in step 3 of Figure 2 seems to be part of the second-last sentence/bullet
Reviewer response
We have added the missing words “continue with the next step”
Occasional Spaces missing between the text and references number at the end of the sentences, for example, personal knowledge(26).
Reviewer response
We have created a space on between the reference and the personal knowledge.
- Careful proofreading and correction with some online grammar apps can improve the quality of the paper
Reviewer response
Thank you, we have done a thorough review and created spaces between a reference and the last word in the statements that are referenced

Reviewer 3 Report
This paper illustrated improvement of health services for TB patients by introducing screening for DM for the TB patients. It reports success story in so doing, which is not unexpected.
The methods used are generally well described. However, the results do not have information on timeline of the changes. Note that the intervention was before covid-19 pandemic but the timing for evaluation was not mentioned. There are certain percentage of facility dropped out and not included in the analysis. They should be analysed to identify characteristics of the centers that fail to complied with the study.
The percentage of the patients undergone DM screening were shown but we have no idea about the denominator.
Author Response
Comments and Suggestions for Authors
This paper illustrated improvement of health services for TB patients by introducing screening for DM for the TB patients. It reports success story in so doing, which is not unexpected.
Reviewer #3 response
We thank the reviewer for taking time to go through the manuscript and provided comments for improving the report. Below, you will find a point-by-point description of how we have attended the comments and suggestion in the manuscript. Original reviewer’s comments are in regular face, responses in red typeface.
The methods used are generally well described. However, the results do not have information on timeline of the changes. Note that the intervention was before covid-19 pandemic but the timing for evaluation was not mentioned. There are certain percentage of facility dropped out and not included in the analysis. They should be analysed to identify characteristics of the centers that fail to complied with the study.
Reviewer response
We appreciate for the comments. The first paragraph of the result section stated “In 2021, 40 health facilities were assessed for their performance of the entire annual year of 2020. Unfortunately, COVID-19 waves interrupted completion of the processes and only 25 (63%) facilities were evaluated.” We state that the implementation of the intervention was done in 2020 and evaluation was performed in 2021. All evaluated sites included information of 2020. We have also stated very clear that we did not assess other facilities because of COVID-19 which resulted into considerable disruption of various implementation of activities.
The percentage of the patients undergone DM screening were shown but we have no idea about the denominator.
Reviewer response
We provided the general denominator in median and IQR for Diabetes and those screened for TB.
The median (IQR) number of individuals receiving DM services in year 2020 at the intervention hospitals was 1760 (309-2570) and control was 98 (30 -6279) This is the denominator
We appreciate very much

Reviewer 4 Report
General comments
Given the situation of double disease burden in Tanzania and other developing countries, there is a high need to integrate co-management NCDs and infectious diseases which have epidemiological co-occurrence and causality. Diabetes mellitus and TB are examples of such diseases hence bidirectional screening and co-management of these diseases can be a promising strategy in averting their incidence and burden in developing countries.
The manuscript is well written in a standard English language The introduction has a logical flow of ideas and concepts. The authors have used rigorous methods and analysis to arrive at meaningful results and findings. The authors have managed to bring up a good discussion reflecting the study findings and their clinical and public health implications.
There are minor issues that the authors should address to improve the manuscript before deciding to publish.
Specific comments
Abstract:
- On page 1, lines 36-37: Mixed methods refers to the use of more than one approach in data collection and analysis e.g. quantitative and qualitative! Is the mixed method refer to the two research designs - cross-sectional and retrospective? I suggest the use of a "cross-sectional retrospective design" instead. Please clarify and amend accordingly.
- On page 2, line 51: Deletes the full stop between "management" and "and"
Background:
- On page 2, lines 73-77: The sentence is unnecessarily too long. Please make two sentences or put a full stop between “preventive” and “countries.”
Methods:
- On page 5, lines 206-207: It is the other way around that hyperglycemia increases the concentration of circulating cytokines. Please revise and amend accordingly and cite the relevant reference.
- On page 6, line 245: Enclose the cited reference “24” with the bracket.
- On page 6, line 264: It seems you have included dispensaries in the category of health centres! In the target site, the dispensaries were mentioned but not seen in the categorization. In which category do dispensaries fall? Please provide the reason.
- On page 7, Figure 2: Replace studies with studies in the sentence “3-doctors immediately went for postgraduate studied.” Also, insert space between the word “for” and “a” in the sentence “1-doctor went fora prolonged maternity leave.”
The manuscript is written in standard English.
Author Response
Comments and Suggestions for Authors
General comments
Given the situation of double disease burden in Tanzania and other developing countries, there is a high need to integrate co-management NCDs and infectious diseases which have epidemiological co-occurrence and causality. Diabetes mellitus and TB are examples of such diseases hence bidirectional screening and co-management of these diseases can be a promising strategy in averting their incidence and burden in developing countries.
The manuscript is well written in a standard English language The introduction has a logical flow of ideas and concepts. The authors have used rigorous methods and analysis to arrive at meaningful results and findings. The authors have managed to bring up a good discussion reflecting the study findings and their clinical and public health implications.
There are minor issues that the authors should address to improve the manuscript before deciding to publish.
Reviewer #4 response
We highly thank the reviewer for applauding favorably this manuscript. We value considerably the time you have spent to review, comments and suggest improvement of this report. Below, you will find a point-by-point description of how the reviewer comments have been attended in the manuscript. Original reviewer’s comments are in regular face, responses in red typeface.
Specific comments
Abstract:
- On page 1, lines 36-37: Mixed methods refers to the use of more than one approach in data collection and analysis e.g. quantitative and qualitative! Is the mixed method refer to the two research designs - cross-sectional and retrospective? I suggest the use of a "cross-sectional retrospective design" instead. Please clarify and amend accordingly.
Reviewer response
We have amended accordingly as you have suggested and now line 36-37 reads “A retrospective and cross-sectional designs were used to access fidelity, acceptability and feasibility of the model”
- On page 2, line 51: Deletes the full stop between "management" and "and"
Reviewer response
The full stop has been deleted and thanks for observation.
Background:
- On page 2, lines 73-77: The sentence is unnecessarily too long. Please make two sentences or put a full stop between “preventive” and “countries.”
Reviewer response
We have amended the sentence and split into two sentences to simplify as follows “The global response toward TB and HIV has led to equally momentous change in recommendations and guidelines such as lipoarabinomannan diagnostic test or novel TB preventive therapy”. “Countries like Tanzania still face systemic challenges on effective delivery of these updated recommendations to front-line healthcare workers with adequate support for implementation (9, 10).
Methods:
- On page 5, lines 206-207: It is the other way around that hyperglycemia increases the concentration of circulating cytokines. Please revise and amend accordingly and cite the relevant reference.
Reviewer response
We are not in the agreement with the reviewers that hyperglycemia increases the concentration of circulating cytokines. We stated that “The algorithm first uses a glucometer, and then to exclude patients with transient hyperglycaemia due to cytokine stimulation (false DM diagnoses) (28), the algorithm used Hb1Ac to confirm the diagnosis and assess the severity of DM. It is clear that Stress hyperglycemia is a result from complex interplay between disturbed cytokine and hormone production, leading to excessive hepatic glucose production and insulin resistance.
- On page 6, line 245: Enclose the cited reference “24” with the bracket.
Reviewer response
We have reflected the reference accordingly.
- On page 6, line 264: It seems you have included dispensaries in the category of health centres! In the target site, the dispensaries were mentioned but not seen in the categorization. In which category do dispensaries fall? Please provide the reason.
Reviewer response
We described the categorization of the health facilities as hospitals if they served the district and above or health centres if they served below the district level. Mean that the health centers and dispensaries were included in one category because they operate below the district level.
- On page 7, Figure 2: Replace studies with studies in the sentence “3-doctors immediately went for postgraduate studied.” Also, insert space between the word “for” and “a” in the sentence “1-doctor went fora prolonged maternity leave.”
Reviewer response
We have amended accordingly and thank you very much for observation
